# Sensitivity and Attachment in an Italian Sample of Hikikomori Adolescents and Young Adults

**DOI:** 10.3390/ijerph20126148

**Published:** 2023-06-16

**Authors:** Alessandra Santona, Francesca Lionetti, Giacomo Tognasso, Chiara Fusco, Graziana Maccagnano, Danila Barreca, Laura Gorla

**Affiliations:** 1Department of Psychology, University of Milano-Bicocca, 20126 Milano, Italy; alessandra.santona@unimib.it (A.S.); g.tognasso@campus.unimib.it (G.T.); c.fusco7@campus.unimib.it (C.F.); g.maccagnano@campus.unimib.it (G.M.); d.barreca1@campus.unimib.it (D.B.); 2Department of Neurosciences, Imagine and Clinical Sciences, University of G. D’Annunzio Chieti-Pescara, 66100 Chieti, Italy; francesca.lionetti@unich.it

**Keywords:** environmental sensitivity, adolescents, attachment, psychological distress, hikikomori

## Abstract

*Hikikomori* is a severe form of social withdrawal increasing among the young Italian population. Hikikomori has been connected to psychological problems and high environmental sensitivity. Nevertheless, only a few studies have been carried out in the Italian context, and they did not analyze several aspects strictly related to the hikikomori phenomenon, such as the role of attachment and sensitivity. We aimed to investigate the relationship between attachment, sensitivity, and psychological problems in a sample of Italian hikikomori. Our sample comprised 72 Italian adolescents and young adults (49 males and 23 females), meanly aged 22.5 years, recruited through online forums and clinical centers for the hikikomori phenomenon. Our participants fulfilled the Highly Sensitive Person Scale (HSPS), the Attachment Style Questionnaire (ASQ), and the Hopkins Symptom Checklist (SCL-90-R). The results showed high psychological issues (i.e., depression and anxiety), environmental sensitivity, and insecure attachment orientations. Moreover, we discovered a significant relationship between attachment dimensions, environmental sensitivity, and psychopathology. Our study sheds light on a novel research path and could help both the researchers and the clinicians who work with people suffering from social withdrawal.

## 1. Introduction

In recent years, the spread and notoriety of hikikomori have increased considerably, attracting interest and attention within the international scientific panorama [1,2].

The Japanese psychiatrist Saito used the term hikikomori for the first time to denote a phenomenon that first appeared in Japan in the late 1990s [3]. It involves prolonged, voluntary self-exclusion in one’s home or room, ranging from months to years. This self-exclusion is not directly caused by psychiatric conditions [1,4,5]. Hikikomori refers not only to a specific psychopathological or medical condition, but also to a complex and multidimensional phenomenon where cultural, sociological, and psychological aspects are strictly connected [6].

The literature on this theme describes two types of hikikomori condition: “primary hikikomori” and “secondary hikikomori”. The first term refers to a condition with no other associated pathologies. In contrast, the second term recalls comorbidity between the hikikomori condition and other psychiatric disorders [7,8].

The most typically associated change in lifestyle of hikikomori sufferers is characterized by the reversal of sleep–wake rhythm and the preference for online social relationships [2,5,6,9].

In 2010, the Japanese government officially defined hikikomori as an active avoidance of social situations characterized by self-isolation in one’s home for at least six months [10]. However, due to the variability of its characteristics and difficulty in detecting cases, there is currently no agreement on the criteria for an official psychiatric diagnosis [11]. Therefore, estimates of its spread are also affected by a similar definitional relativism, with numbers ranging from 540,000 to a million cases recorded in Japan. At the same time, the list of countries with variable prevalence rates according to the different criteria adopted outside the Japanese context is long. For instance, the prevalence of hikikomori is 2.3% in Korea; 1.9% in China, particularly in Hong Kong; and 12.6% in Spain and France [1,11,12,13,14].

Even in Italy, although there are no specific epidemiological studies; in 2013 there were an estimated 100,000 to 240,000 cases of social withdrawal through surveys from the National Federation of the Orders of Surgeons and Dentists (FNOMCeO) and the Minotauro Institute of Milan [15].

Crepaldi [16] conducted the first survey on Italian parents and children self-assessed as hikikomori through an online questionnaire. The survey involved 288 parents and 89 hikikomori sufferers who were predominantly males with an average age of 23 years, with social withdrawal behavior manifesting around 15 years. Half of the participants perceived their relationship with their parents as bad or poor, and a percentage of 65% of hikikomori would seek expert help. Parents reported that 63.9% of sons had an altered sleep-wake cycle and showed depressed mood and apathy. 

### 1.1. Attachment and Hikikomori

When studying the hikikomori phenomenon, scholars have also focused on the first meaningful relationships experienced, such as those with caregivers. They have hypothesized that one of the possible reasons behind the voluntary self-exclusion of the hikikomori syndrome could be related to these relationships [17,18]. 

According to Bowlby’s Theory of Attachment [19], the human being is endowed, from birth, with a biologically determined system aimed at maintaining the physical and emotional closeness to the figure of the caregiver through typical attachment behaviors (i.e., weeping and smiling) [20].

Repeated childhood experiences with attachment figures lead to the development of internal working models (IWM), defined as internalized representations of the self, others, and relationships between the self and others. If a child constructs a secure attachment with their parents in childhood, they develop an idea of themselves as good enough to be loved and respected by others. Instead, if they build an insecure attachment with their parents in childhood, they are more likely to worry about abandonment, experience high anxiety levels while in a relationship with others, and exaggerate their needs. They could also fear interpersonal closeness, avoid significant relationships with others, or rely primarily on themselves [21]. Even in adulthood, IWMs contribute to emotional and behavioral regulation in affective relationships with romantic partners and peers. Indeed, individual variations of relational anxiety and avoidance in intimate relationships reflect the underlying strategies of overactivation and deactivation of the adult attachment behavioral system [22,23].

Regarding the attachment developed by hikikomori, Hattori [23] hypothesized the insecurity in the attachment system of people suffering from the hikikomori condition. This could be caused by parental indifference and insensitivity towards their children. As a result of this insecurity, children fear their parents and find it difficult to express their feelings to them. Thus, the fear of being rejected by their parents drives hikikomori to hide their true identity. This mode ends in the development of an attachment relationship characterized by patterns of insecurity. Despite the relevance of IWMs and insecure attachment patterns for later significant relationships in adulthood, adult attachment orientations in hikikomori sufferers still appear scarcely investigated in the psychological literature. 

The presence of insecure attachment in hikikomori sufferers has been explored by the literature referring to the Japanese practice of *Mushi* (emotional coldness) and *amae*, which indicates a child’s demand for the absolute dependence from the mother anticipating needs and encouraging the symbiotic union itself, aimed at achieving a fundamental connection and social belonging. Therefore, to avoid the creation of a symbiotic bond with their children, Japanese mothers transmit a kind of emotional coldness to them, ignoring them [1,4,10,18,24].

The impact that *amae* can have on the emotional self-regulation of the child could imply the impossibility of experiencing the frustration of one’s needs, with potential exposure to situations of failure that are not easy to manage. *Amae* could also involve a feeling of shame or embarrassment, and withdrawing from the challenge may appear to be the only solution to avoid similar emotions. The conceptual affinity between the *amae* and the Attachment’s construct has led some scholars to study the first significant relationships in depth. 

In addition, the idea of *amae*, according to the literature, has a conceptual affinity with the Western attachment construct as they both develop towards the end of the first year of life and undergo several changes in the following years. Other similar characteristics between the attachment construct and the *amae* are that both are stimulated by the caregiver’s responsiveness and empathy, and allow for the development of emotional competence [17,18].

As the child grows up, individuals’ attachment styles’ characteristics strongly influence how social interactions are built and how peer relationships (friendships and romantic relationships) are experienced [25]. Researchers [26] have conceptualized individual differences in adult attachment in terms of different functional organizations of the attachment system, which can be reflected in individuals’ affective and behavioral regulation processes in intimate interpersonal relationships. Specifically, individuals appear to differ based on different relational orientations defined by abandonment anxiety and interpersonal relationship avoidance. These dimensions result from the interplay between the behavioral attachment system [19] and one’s own relational attachment experiences with significant others [27]. Thus, abandonment anxiety and interpersonal relationship avoidance have been conceptualized as dimensions founding attachment patterns and regulating the activation of the attachment system toward romantic partners or intimate friends in adulthood [22,23,26].

The theme of emotional self-regulation and the first meaningful relationships allows for the exploration of the quality of attachment most frequently associated with hikikomori, with a subsequent prevalence of insecure attachment patterns being discovered [18,24]. In particular, Krieg and Dickie [18] have proposed a psychosocial evolutionary model linked to the phenomenon of social withdrawal, where the presence of insecure-ambivalent attachment, peer rejection, and shyness contribute to the development of the condition of hikikomori [18,28]. However, adult attachment has been scarcely explored in people with hikikomori, despite the potential relevance of interpersonal-based affective and behavioral regulation in hikikomori syndrome.

### 1.2. Environmental Sensitivity and Hikikomori

Sensory processing sensitivity is a human trait that consists of the ability to perceive and process information about the environment. This personality trait concerns a tendency to process sensory information more deeply than others [29]. For this reason, the literature defines highly sensitive persons (HSP or PAS, in Italian and Spanish) as people who tend more than others to pay attention to the details around them, comparing what they perceive in the present with similar experiences they had in the past.

The personality trait of sensitivity has different values according to the situation; if a person is having a positive experience, being sensitive to details could increase their well-being. On the other hand, if a person faces a difficult or negative situation, the increased receptivity to stimuli can become disadvantageous [30].

For this reason, sensitivity is related to several aspects, described with the acronym “D. O. E. S.” [31]:-“Depth” for the depth with which the processing takes place.-“Overstimulation” is a perceived overload of stimuli, leading to behavioral inhibition, a need to reflect, check, and observe before acting.-“Emotional responsiveness/empathy” is the greater emotional reactivity to positive images and the greater empathy compared with non-HSP people.-“Sensitive to subtle stimuli” reflects the perception of environmental and social details not perceptible by non-HSP. This does not depend on the presence of more acute senses from the perceptual point of view, but essentially concerns the much more careful processing of information.

A high sensitivity brings highly sensitive persons to assess the consequences of future actions and predictions, but, at the same time, there is a lower threshold for the central nervous system (CNS), which is overloaded compared with non-HSP when the environment is more chaotic [31,32].

Without suffering from sensory processing problems, they tire quickly and are more likely to experience stress and feel overwhelmed due to the over-stimulation they perceive. In addition, constant overstimulation can also encourage people to avoid social situations and stimuli [29,31,33]. Indeed, one study hypothesized that hikikomori could be considered highly sensitive people because they perceive the environment as highly critical and judgmental towards themselves and, for this reason, try to increasingly avoid it [18].

Krieg and Dickie [18] explained that temperamental and dispositional characteristics, such as high sensitivity, shyness, and irritability, are associated with ambivalent attachment and parental rejection in hikikomori. The latter aspects are notoriously implicated in developing psychological difficulties, including social withdrawal [18,34].

At first, scholars divided people into very sensitive and insensitive categories [29]. However, recent studies [29] have shown the existence of a third category of people with medium sensitivity, reorganizing the population into three groups: people with high, medium, and low sensitivity. However, the authors suggest defining sensitivity as a continuum where people are predominantly placed within three groups. In this continuum, HSP people represent the minority [29].

HSP is often associated with severe psychological disorders such as anxiety and depression, especially if there are negative past experiences, as the latter induces less control over emotional reactions. The three types of sensibility identified differ in neuroticism, emotional reactivity, and extraversion; accordingly, HSPs score significantly lower in the latter and higher in the first two compared with individuals with medium and low sensitivity [29,30]. 

Similar developments have allowed us to conduct our investigation to verify, among other hypotheses, the HSP trait’s actual presence in our sample of hikikomori [29,31,33].

### 1.3. Psychiatric Symptoms

The literature on social withdrawal reports that when people start to distance themselves from their everyday and real lives, they initially experience a feeling of well-being as they escape from a reality perceived as painful. However, prolonged withdrawal inevitably led to the emergence of aspects worthy of clinical attention [1].

It has recently been proposed to consider the presence of psychiatric symptoms to assess the severity of isolation in hikikomori [34]. Specifically, three main criteria have been proposed:-Marked isolation in one’s own home.-At least six months of self-reclusion.-Significant impairment of functioning and perceived discomfort related to isolation.

Moreover, the literature has found that some diagnoses are more associated with the hikikomori syndrome than others, paying particular attention to the riskier cases who might attempt suicide [1,35,36,37].

Some clinical manifestations related to the hikikomori phenomenon are avoidant personality disorder, where social inhibition, a sense of inadequacy, and hypersensitivity to criticism coexist; post-traumatic stress disorder (PTSD), determined by an intrusive memory of painful events; and autism, because of its communication difficulties [38]. Studies have also highlighted the presence of high levels of anxiety in people suffering from social withdrawal [1,12]. In addition, hikikomori sufferers tend to mainly experience depression, pervasive sadness and a sense of emptiness, and social phobia [6,9,36,39].

The psychological literature has highlighted the relationship between depressive and anxious symptoms and hikikomori severity [40]. Moreover, assuming withdrawal from live interpersonal relationships as a critical feature of the syndrome, it is crucial to investigate the role assumed by personal sensitivity related to hikikomori syndrome. Indeed, interpersonal sensitivity has been defined as the accuracy of interpreting the meaning of nonverbal cues indicative of emotion, roles, relationships, deception, and personality [41]. Along this line of research, a recent study [42] referred to anxiety, depression, and interpersonal sensitivity as relevant psychiatric dimensions in understanding hikikomori syndrome in clinical and non-clinical samples. Although withdrawal may be influenced by the ability to interpret nonverbal cues, interpersonal sensitivity needs further comprehension in hikikomori sufferers, alongside more well-established symptomatology such as depression and anxiety.

### 1.4. Aims and Hypothesis

The current study explores the relationship between attachment, environmental sensitivity, and psychological distress in a sample of Italian hikikomori. 

As far as we know, few studies have been conducted on the hikikomori phenomenon in Italy, especially with a focus on adult attachment, sensitivity, and psychopathology (in terms of interpersonal sensitivity, depression, and anxiety). 

According to the previous studies, we expected the following:(a)There would have been a relationship between specific aspects of attachment to romantic partners and intimate peers, environmental sensitivity, and psychopathology (interpersonal sensitivity, depression, and anxiety).(b)Higher levels of anxious and avoidant attachment dimensions would have been connected to high interpersonal sensitivity, depression, and anxiety, while secure attachment would have been a protective factor by decreasing psychological depression, anxiety, and interpersonal sensitivity.(c)High levels of environmental sensitivity would have been related to high psychopathological interpersonal sensitivity, depression, and anxiety.

## 2. Materials and Methods

### 2.1. Participants

The survey involved 72 volunteer participants, including 49 males (68.1%) and 23 females (31.9%), who identified themselves with the hikikomori condition after reading a brief description of the hikikomori phenomenon. The participants’ ages varied between 12 and 33 years, averaging 22.5 years (SD = 4.7). To better analyze the sample’s characteristics, we decided to divide the participants’ ages into two categories: the first includes the youngest participants, aged from 12 to 17 years old (*N* = 12), while the second consists of the most senior participants, aged from 18 to 33 years old (*N* = 85). Most participants (77.8%, *N* = 56) were single, while 22.2% (*N* = 16) were in a relationship. As for the educational level, participants mainly had a diploma from high school (51.4%, *N* = 37), followed by a middle school degree (38.9 %, *N* = 28), and a bachelor’s degree (9.7%, *N* = 7). Finally, as for working activities, the majority (47.2%, *N* = 34) were students, followed by 37.5% (*N* = 27) who did not have a job, 9.7 % (*N* = 7) who were working, and 5.6% (*N* = 4) who were both studying and working at the same time.

### 2.2. Procedure

We collected data between September 2019 and February 2020.

Research participants were recruited primarily online through Italian hikikomori groups and forums, whose administrators were contacted in advance to express their consent regarding data collection within their groups and forums. In addition, some participants were approached at specialized centers for hikikomori located in Italy, who, likewise, agreed to participate in the project.

We followed the provisions of Italian law 196/2003 to collect the participant’s consent to complete the questionnaire. Before starting the questionnaire, the participants read a brief explanation about the content and purpose of the study. The Ethics Committee of the psychology department of Milano-Bicocca University previously approved the research project. As some participants were underage, we administered two informed consents, one to the underage participants and a second to their parents. According to the ethical guidelines for research in psychology for underage participants, we adapted the informed consent to make it accessible and comprehensible and ensure their free decision to participate in the study.

We followed two types of questionnaire administration: online using Qualtrics and paper-and-pencil when possible.

### 2.3. Measures

Our sample fulfilled the following instruments:

*Ad hoc questionnaire:* We created a biographic questionnaire consisting of 27 questions to collect demographic information (such as year of birth, gender, educational qualification, profession, and marital status) and aspects the most recent studies consider specific to a Hikikomori person. In particular, we explored the educational qualification and work of the parents, the financial maintenance and the cohabitation or not with the family of origin, as well as the lifestyle adopted in the last six months, concerning sleep–wake rhythm (e.g., “I prefer to sleep during the day”), the frequency with which one leaves home (e.g., “I spend most of my time at home”), the preference of face-to-face relationships rather than online, the avoidance of social situations (e.g., “I tend to avoid going to school/work”), and eating habits and the use of technology (e.g., “I spend most of my time on the computer”).

*Attachment Style Questionnaire:* We used the Italian version of the Attachment Style Questionnaire (ASQ) [43] to evaluate adult attachment and analyze the attachment style. The original version [20] comprised 65 items, but 25 were discarded, leading to the current and final version of 40 items. ASQ can be used with adolescents and adults as it identifies individual differences in personal and romantic attachment style, even in the absence of romantic relationships.

ASQ has five scales: confidence (ASQ-C), discomfort with closeness (ASQ-DC), need for approval (ASQ-NA), preoccupation with relationships (ASQ-PR), and relationships as secondary (ASQ-RS). Each item is rated on a six-point scale, ranging from 1 (totally disagree) to 6 (totally agree), and all of the scales can be grouped into the two latent dimensions of anxiety and avoidance. Moreover, ASQ detects different types of attachment styles labeled as secure, avoidant, anxious/ambivalent, and fearful, but the authors suggested not considering attachment as divided into categories but as a unique dimension.

We calculated Cronbach’s alpha for all of the subscales, and our results showed adequate internal consistency. In particular, Cronbach’s alpha was 0.699 for confidence, 0.727 for discomfort with closeness, 0.696 for need for approval, 0.751 for preoccupation with relationships, and 0.771 for relationships as secondary.

In this paper, we considered ASQ-C as an index of the secure attachment style and labeled it ASQ-SECURE (Cronbach’s alpha = 0.699). We also summed ASQ-DC and ASQ-RS to create a unique avoidant attachment style score labeled ASQ-AVOIDANT (Cronbach’s alpha = 0.782). Finally, we summed ASQ-NA and ASQ-PR to create a unique index of the anxious attachment style labeled ASQ-ANXIOUS (Cronbach’s alpha = 0.794).

*Hopkins Symptom Checklist:* We used the Italian version of the Hopkins Symptom Checklist (SCL-90-R) [44] to evaluate psychological distress by focusing on several dimensions. The instrument was initially developed as a discomfort scale by Parloff and colleagues [45] to assess improvements in patients suffering from neurotic disorders.

The Italian version, as the original, is formed by 90 items that assess nine dimensions of psychological distress: somatization (physical discomfort), obsessive–compulsive (the presence of obsessive thoughts, impulses, and actions), interpersonal sensitivity (e.g., feelings of inadequacy), depression (depressive symptoms), anxiety (anxiety symptoms), anger–hostility (the presence of aggressive actions), phobic anxiety (persistent fear towards situations, objects, and people), paranoid ideation (deliriums or bizarre thoughts), and psychoticism (e.g., schizophrenia).

The participants were asked to report how much they felt distressed or bothered by a particular situation during the past week on a scale where 0 is “not at all”, and 4 is “extremely”.

For each scale, scores were categorized from light severity (with a score of <45) to extreme severity (with a score between 65 and 75). The instrument also had a total score (range 90–450) calculated by summing all of the subscales’ scores.

In line with previous research [42], we focused on three subscales (depression, anxiety, and interpersonal sensitivity).

Finally, we obtained a high level of internal consistency in all of the subscales we were interested in, with a Cronbach’s alpha of 0.897 for depression, 0.889 for anxiety, 0.819 for interpersonal sensitivity, and 0.974 for the Global Severity Index.

*Highly Sensitive Person Scale:* We used the Italian version of the Highly Sensitive Person Scale (HSPS) [29,46] to evaluate the sensory-processing sensitivity, which is conceptualized as sensitivity to subtle stimuli and being over-aroused by stimuli coming from the environment. Originally composed of 27 items, HSPS is a self-report of 12 items assessed on a 7-point Likert scale. The instrument is composed of three subscales: ease of excitation (EOE), which is being easily overwhelmed by stimuli both internal and external; aesthetic sensitivity (AES), which is capturing aesthetic awareness; and low sensory threshold (LST), which is unpleasant sensory arousal to external stimuli. In addition to these subscales, HSPS has a total score obtained by taking the average across all 12 items. High scores represent high levels of sensitivity.

Finally, in our sample, Cronbach’s alpha was 0.810 for ease of excitation, 0.741 for aesthetic sensitivity, 0.383 for low sensory threshold, and 0.699 for the total score. Cronbach’s alpha showed adequate internal consistency in all of the dimensions, except for the low sensory threshold, which showed low reliability that should be considered when interpreting the results.

### 2.4. Analysis Plan

Data analysis was conducted using Jamovi 2.3.28 statistical software (The Jamovi project, Sydney, Australia). The normality of the distribution of the variables examined was tested using the Shapiro–Wilk test, which produced significant results in several cases. For this reason, we decided to use non-parametric tests as they do not require a sample drawn from a normally distributed population. Moreover, as our participants expressed high levels of psychological distress, we categorized their answers in binary variables, where 0 corresponds to low or medium levels of psychological distress and 1 corresponds to high or extremely high levels of psychological distress. To categorize participants’ answers, we used the Hopkins Symptom Checklist (SCL-90-R) scale’s clinical cut-off: a score between 45 and 54 was categorized as a low or medium level of psychological distress, while a score between 55 and 75 was a high or extremely high level.

Firstly, descriptive analyses were conducted to understand the typical dimensions involved in the hikikomori condition and attachment, sensitivity, and psychological distress in our sample. Secondly, we performed Kendall’s Tau-b correlations and logistic regressions with Bonferroni correction to explore the relationship between attachment, environmental sensitivity, and psychopathological aspects, which we were interested in (depression, anxiety, interpersonal sensitivity, and global index).

## 3. Results

### 3.1. Preliminary Analysis 

#### 3.1.1. Dimensions Involved in the Hikikomori Phenomenon

To understand whether our participants presented some of the most common characteristics associated with the Hikikomori phenomenon, we investigated the time spent at home, the avoidance of social interactions, the tendency to have online relationships, the frequency of time spent sleeping during the day, the amount of time spent in the bedroom and, finally, hanging out with friends. Our participants were asked to range their habits from always to never. 

Table 1 and Table 2 report the descriptive analyses.

#### 3.1.2. Correlations and Logistic Regressions

Table 3 reports the correlations for the study variables. Attachment security was negatively correlated with interpersonal sensitivity and anxiety, while attachment avoidance was positively correlated with interpersonal sensitivity and depression. Furthermore, attachment anxiety was positively associated with interpersonal sensitivity and depression. As for environmental sensitivity, we found a significant positive correlation between ease of excitation and all the psychopathological dimensions. Finally, significant positive correlations between low sensory threshold, depression, and anxiety were found.

We then performed binomial logistic regressions using interpersonal sensitivity, depression, and anxiety as dependent variables, and attachment-related categories and environmental sensitivity dimensions as independent variables. The model also included the age and gender of our participants as control variables, which showed no significant effects for all of the dependent variables.

Table 4 and Table 5 show measures of goodness of fit for the three models and binomial logistic regressions’ results.

Our results show that participants with anxious attachment are more likely to experience high or extremely high levels of depression (OR = 1.17, B = 0.16, *p* = 0.009) or interpersonal sensitivity (OR = 1.09, B = 0.09, *p* = 0.03). We did not find significant relations between secure attachment and psychological distress and between attachment and anxiety.

As for the environmental sensitivity, our results revealed a significant relationship between ease of excitation and interpersonal sensitivity: being easily overwhelmed by internal and external stimuli increases the odds of experiencing severe forms of interpersonal sensitivity (OR = 2.93, B = 1.07, *p* = 0.02). 

Moreover, a low sensory threshold significantly impacted depression and anxiety: participants with unpleasant sensory arousal to external stimuli were more likely to express high levels of depression (OR = 2.44, B = 0.89, *p* = 0.018) and anxiety (OR = 2.1, B = 0.78, *p* = 0.009) as psychological distress dimensions.

We did not find significant relations between the total score of environmental sensitivity and all the psychological distress dimensions.

## 4. Discussion

The present study aimed to explore the relationship between attachment to romantic partners and intimate peers, environmental sensitivity, and psychopathology (regarding depression, anxiety, and interpersonal sensitivity) in a sample of Italian hikikomori.

To the best of our knowledge, no studies have previously focused on the interplay between adult attachment dimensions and environmental sensitivity with psychopathological issues in a sample of Italian hikikomori sufferers. 

Overall, our main findings suggest the presence of high levels of clinical symptomatology in the sample. Moreover, our results show the relevant role of attachment anxiety and avoidance orientations on different psychopathological dimensions, such as depression and interpersonal sensitivity. Furthermore, environmental sensitivity was confirmed as a critical personality trait in understanding hikikomori syndrome.

Concerning our first research question, our hypothesis was supported as we found significant associations between psychopathology and attachment to significant ones. Higher levels of attachment security were significantly associated with lower anxious symptomatology and interpersonal sensitivity. Furthermore, higher levels of attachment dimensions of abandonment anxiety and interpersonal relationship avoidance were significantly related to increased depression and interpersonal sensitivity symptomatology. These results are consistent with previous ones [21,23] stating that secure patterns in attachment to partners and intimate friends (or low levels in the dimensions of anxiety toward abandonment and avoidance of close interpersonal relationships) would be a protective factor against the emergence of clinically significant psychological distress, as it is usually related to the use of constructive and effective affect-regulation strategies.

As for environmental sensitivity, we found that high levels of ease of excitation were significantly associated with heightened levels of interpersonal sensitivity. Concurrently, a low sensory threshold was related to increased levels of depressive and anxious symptomatology in our sample. In turn, aesthetic sensitivity, which captures aesthetic awareness, did not display any significant association with psychopathological issues.

These findings may open new research directions. As far as we know, the role of attachment relationships to significant others in adulthood and environmental sensitivity has been overlooked in the psychological literature regarding the mental health of hikikomori sufferers. Both these aspects comprise significant interpersonal dimensions related to psychopathological issues possibly prone to emerge in conditions of prolonged withdrawal. Our results seem to confirm the relevance of studying their relationship with hikikomori syndrome.

Concerning our second research questions, our hypotheses were partially confirmed, as we did not find security in adult attachment as a protective factor against the emergence of psychopathology. However, we discovered that participants with high levels in attachment dimensions of relationship avoidance or abandonment anxiety were likelier to experience high or extremely high levels of depression or interpersonal sensitivity. These results are consistent with the literature highlighting how insecure orientations in attachment patterns can affect the outcomes of hikikomori sufferers regarding psychological maladjustment [22]. Considering the attachment system as a motivational system that contributes through interpersonal strategies to emotion regulation during the whole life span [19,20], we can interpret these results by hypothesizing that hikikomori sufferers may be adversely impacted by prolonged social withdrawal and that this may contribute to the failure of interpersonal regulation strategies. These circumstances could emphasize the perception of overstimulation and depressive symptomatology in hikikomori sufferers, as found in our study. Alternatively, we could consider individual differences in attachment dimensions in terms of different affective and behavioral regulation processes in intimate relationships [26]. Assuming this, it is also possible that hikikomori sufferers with anxious or avoidant attachment orientations may experience interpersonal relationships as highly stressful and thus be reinforced in their withdrawal behaviors. Considering individuals’ differences in attachment patterns in adulthood as the product of repeated activations of the attachment system in relationships with significant others throughout the life span [19,27], it seems relevant to further understand whether insecure attachment patterns found in hikikomori infancy [23] may ultimately impact relational patterns toward significant peers in adulthood and contribute to withdrawal behaviors.

Contrary to what we expected, no significant associations were observed between adult attachment dimensions and anxious psychopathology or between attachment security and symptoms of psychopathology in our sample. These findings do not align with previous research reporting significant relationships between these aspects [23,47]. A possible explanation is that our study was underpowered and we were unable to detect it because of the small sample’s dimensions, which did not permit us to detect small-to-medium effects.

Concerning our third research question, regarding environmental sensitivity, in line with previous literature [29,31,33], our study confirmed the HSP trait’s actual presence in hikikomori. Indeed, hikikomori sufferers could have an increased receptivity to internal or external stimuli when facing a difficult or negative situation, an aspect that can become disadvantageous [30].

Specifically, our study highlighted the role of two key dimensions (the ease of excitation and the low sensory threshold) of environmental sensitivity in the experience of hikikomori syndrome. Participants who tended to be easily overstimulated by internal and external stimuli were more likely to experience severe interpersonal sensitivity. In this sense, it is possible to identify a vicious cycle for HSP individuals with hikikomori syndrome. The more they withdraw from social interactions and external stimuli, the more they may experience highly challenging and overstimulating interpersonal relations, reinforcing their need for isolation and behavioral inhibition.

Finally, our results show that a low sensory threshold seemed to have a significant probability of increasing anxiety and depression participants with unpleasant sensory arousal to external stimuli were more likely to express high anxiety and depression levels. Although the low sensory threshold could be present in hikikomori sufferers even before the onset of social isolation [18], it could be exacerbated by withdrawal behaviors and prolonged lack of exposure to external and less controllable stimuli. As the withdrawal period increases, the low sensory threshold could further decrease, contributing to the perception of external stimuli as overwhelming and a source of concern in hikikomori sufferers. Previous literature [48] showed that high environmental sensitivity correlates with internalizing symptoms, such as anxiety and depression, in adult samples. Assuming environmental sensitivity and its features, as temperamental dispositions, that interact meaningfully with environmental factors [48], it is conceivable that prolonged withdrawal could lead to increased levels of internalizing symptoms, thus reinforcing isolation behaviors in hikikomori sufferers.

This study has some limitations that should be considered when interpreting the results. First, as the study had a cross-sectional design, it is impossible to make causal inferences. Second, our results cannot be widely generalized due to the recruitment methods and the small sample size. Third, due to the data collection method (self-report questionnaires), social desirability response bias could also have affected the study results. Finally, the low value of Cronbach’s alpha in the low sensory threshold cautions us from generalizing results concerning this dimension of environmental sensitivity.

Despite these limitations, this was the first study to highlight the importance of exploring how adult attachment and environmental sensitivity can be related to psychopathology expressions in hikikomori sufferers, highlighting two different paths. The first suggests that insecure attachment orientations to romantic partners or significant peers, namely high abandonment anxiety and interpersonal relationship avoidance, may impact depression and interpersonal sensitivity. The second path underlined how some HSP traits, i.e., the ease of excitation and low sensory threshold, possibly relate to psychopathological issues in hikikomori syndrome and deserve further attention.

It is interesting to note that the current study was conducted before the COVID-19 outbreak. As participants in the current study experienced high levels of psychological distress, it could be possible that the COVID-19 pandemic worsened this aspect. Indeed, several studies showed that the global pandemic significantly impacted the mental health of adolescents and young adults, so we could imagine that suffering from the hikikomori condition would have been a risk factor during the pandemic. Future studies should explore this theme by focusing on the consequences of the COVID-19 pandemic on hikikomori sufferers.

Future research should also explore these themes using larger samples and focusing on the mechanisms underlying the relationship between attachment to romantic partners or close peers, environmental sensitivity, and psychological distress. Furthermore, specific directions of influence in the relationship between adult attachment avoidance of interpersonal relationships and anxiety of abandonment, social withdrawal, and psychopathological outcomes should also be investigated.

Our findings suggest that future studies could focus on possible practical applications of these findings for clinical work with young people suffering from hikikomori syndrome. It might be helpful to explore how HSP traits—such as low sensory threshold, ease of excitability, and insecure attachment orientations—are associated with clinical features of psychopathology to enhance psychological well-being in hikikomori sufferers.

## 5. Conclusions

We examined attachment to romantic partners and significant peers, environmental sensitivity, and psychological distress in adolescents and young adults with hikikomori syndrome. In particular, we focused on the relations between environmental sensitivity, attachment and interpersonal sensitivity, anxiety, and depression expressed by hikikomori sufferers.

The current study sheds light on the hikikomori phenomenon, which is relatively recent in the Italian context, by exploring its characteristics and psychological aspects.

Our findings allow us to draw important conclusions. First, we discovered that our sample expressed high and extreme levels of interpersonal sensitivity, depression, and anxiety. Second, we found out that an anxious attachment increased the odds of experiencing extremely high levels of interpersonal sensitivity and depression, while there were no significant connections between the other two dimensions of attachment (secure and avoidant) and psychopathological aspects.

Finally, we discovered that the ease of excitation and the low sensory threshold were significantly linked to higher psychological distress.

The knowledge developed and the concepts presented in this study can help researchers and clinical psychologists work with adolescents and young adults with hikikomori syndrome.

## Figures and Tables

**Table 1 ijerph-20-06148-t001:** Participants’ characteristics connected to the hikikomori phenomenon (*N* = 72).

*Habits*	Time Spent Home	Avoidance Social Interactions	Tendency to Have Online-Relationships	Frequency of Sleeping during the Day	Frequency of Staying in the Bedroom	Hanging out with Friends
Always	*N* = 30 41.7%	*N* = 12 16.7%	*N* = 10 13.9%	*N* = 9 12.5%	*N* = 23 31.9%	*N* = 2 2.8%
Often	*N* = 31 43.1%	*N* = 37 51.4%	*N* = 18 25%	*N* = 19 26.4%	*N* = 28 38.9%	*N* = 12 16.7%
Sometimes	*N* = 10 13.9%	*N* = 21 29.2%	*N* = 30 41.7 %	*N* = 17 23.6%	*N* = 19 26.4%	*N* = 29 40.3%
Never	*N* = 1 1.4%	*N* = 2 2.8%	*N* = 14 19.4%	*N* = 27 37.5%	*N* = 2 2.8%	*N* = 29 40.3%

**Table 2 ijerph-20-06148-t002:** Descriptive analyses for attachment, environmental sensitivity, and psychological distress.

	Mean	SD
ASQ—secure	21.6	5.99
ASQ—avoidance	67.7	10.8
ASQ—anxious	64.7	11
Ease of excitation (EOE)	5.79	1.08
Aesthetic sensitivity (AES)	5.50	1.21
Low sensory threshold (LST)	4.14	1.14
Highly Sensitive Person Scale—total (TOT-HSP)	62.6	9.02
	Low/Medium Severity	High/Extreme Severity
	N (%)	N (%)
Interpersonal sensitivity (INT)	19 (26%)	53 (74%)
Depression (DEP)	16 (22%)	56 (78%)
Anxiety (ANX)	28 (39%)	44 (61%)

**Table 3 ijerph-20-06148-t003:** Tau-b bivariate correlations.

	Interpersonal Sensitivity (INT)	Depression (DEP)	Anxiety (ANX)
ASQ—secure	−0.211 *	−0.096	−0.223 *
ASQ—avoidant	0.215 *	0.279 **	0.162
ASQ—anxious	0.327 **	0.305 **	0.153
Ease of excitation (EOE)	0.376 ***	0.232 *	0.298 **
Aesthetic sensitivity (AES)	−0.097	−0.074	−0.032
Low sensory threshold (LST)	0.125	0.213 *	0.340 **
Highly Sensitive Person Scale—total (TOT-HSP)	0.200 *	0.135	0.207

* *p* < 0.05; ** *p* < 0.01; *** *p* < 0.001.

**Table 4 ijerph-20-06148-t004:** Goodness of fit.

	INT	DEP	ANX
Χ^2^ (gdl)	24.7 (8) **	24.6 (8) **	17.9 (8) *
R^2^ McF	0.312	0.336	0.199
R^2^ N	0.447	0.465	0.319

* *p* < 0.05; ** *p* < 0.01.

**Table 5 ijerph-20-06148-t005:** Binomial logistic regression.

	INT			DEP			ANX		
	B (SE)	OR	*p*	B (SE)	OR	*p*	B (SE)	OR	*p*
Intercept	−14.36 (6.71)	5.79	0.03	−18.21 (7.5)	1.23	0.01	−5.23 (4.5)	0.00	0.25
Gender	−0.35 (0.87)	0.701	0.68	−0.81 (0.95)	0.442	0.85	−0.24 (0.67)	0.779	0.71
Age	0.12 (0.08)	1.12	0.14	−0.01 (0.08)	0.987	0.87	0.04 (0.06)	1.04	0.49
ASQ—secure	−0.00 (0.07)	0.99	0.95	0.02 (0.07)	1.02	0.74	−0.065 (0.03)	0.93	0.26
ASQ—avoidant	0.04 (0.04)	1.04	0.26	0.08 (0.04)	1.09	0.07	0.016 (0.03)	1.01	0.62
ASQ—anxious	0.09 (0.04)	1.09	**0.03**	0.16 (0.06)	1.175	**0.009**	0.029 (0.03)	1.03	0.34
EOE	1.07 (0.48)	2.93	**0.02**	−0.02 (0.42)	0.972	0.94	0.176 (0.32)	1.19	0.59
AES	−0.55 (0.40)	0.57	0.16	0.07 (0.42)	1.08	0.85	−0.20 (0.31)	0.81	0.50
LST	0.23 (0.34)	1.26	0.49	0.89 (0.37)	2.44	**0.018**	0.78 (0.30)	2.1	**0.009**
TOT-HSP	0.05 (0.03)	0.10	0.15	0.05 (0.04)	1.05	0.242	0.04 (0.03)	1.05	0.136

Note: **B** is the estimated parameter for the independent variable; **SE** is the standard error of B; **OR** stands for odds ratio, which is the probability of occurrence of the event divided by the probability of not occurrence the event; ***p*** is the *p*-value (significant for *p* < 0.05).

## Data Availability

The data presented in this study are available upon request from the corresponding author.

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
