# Peer review of "Sensitivity and Attachment in an Italian Sample of Hikikomori Adolescents and Young Adults"

_ijerph, 2023, doi:10.3390/ijerph20126148_

Round 1

Reviewer 1 Report

Dear authors,

I found your work really interesting and well conducted from a methodological point of view. The constructs are well argued and the introduction part is well in-depth. I also appreciated the way the results were discussed and the quality of the writing. I would only suggest to better explaining hypotheses a and b, for example  mentioning which psychopathological aspects you refer to (b) and by c

Author Response

Dear reviewer,

We thank you for your comment and are glad you found our work interesting. We changed the hypotheses by better clarifying them, following your suggestion. In particular, we mentioned the psychopathological aspects (interpersonal sensitivity, depression, and anxiety) and a more specific hypothesis regarding environmental sensitivity and all psychopathological aspects. You can find these amendments on page 5 of the manuscript.

Reviewer 2 Report

Introduction until Aims and Hypothesis.

The article is written well; however, the authors imply readers are very experienced at the statistical level. Since hikikomori is little studied but is a pretty important topic clinically, I suggest being less synthetic in the statistical part.

Page 6. Attachment Style Questionnaire.

In the description of the ASQ, correct "ASQRS" in "ASQ-RS" as was done in the rest of the article.

Page 7. 2.4 Analysis Plan

Jamovi's version always has a third number (e.g., 2.3.0). You may have used multiple versions (e.g., 2.3.22 and 2.3.26). If so, I suggest indicating, "Data analysis was conducted using the Jamovi statistical software in multiple versions 2.3.X." 

The test is "Shapiro"-Wilk (SHA), not Saphiro-Wilk.

If you use Kendall's Tau-B correlations, why use metric analyzes after?

In Table 1, it needs to be clarified what is in the first column. I would suggest moving "n = 72" to the caption and using "Habits" as a column title; in fact, I believe that the content of the column depends on "asked to range their habits from always to never." In addition, in Table 2 (where you have space), add the relative words to EOE, AES, and LST.

In the explanatory text of Table 3 (but also in all results), I suggest writing the names of the variables used with the capital initial (e.g., "Interpersonal sensitivity") to help connect the text with the contents of the tables.

I suggest indicating the type of correlation (Tau-b) in the table's title.

In Table 3, I suggest adding the size (as in Table 2) and expanding the names of EOE, AES, and LST. Finally, add a legend for INT, DEP, and ANX.

Pages 9-10.

I understand you have made three logistical regressions, one for each dependent variable. Have you considered also using Bonferroni's correction (alfa/3)?

In logistical regression, you use both age and gender, with non-significant results. It would be appropriate to indicate at least the probability of these non-significant results (for future meta-analysis) and specify if they remained in the model.

I also suggest separating the two tables that are currently Table 4.

In addition, only some of the article's readers know the logistical regression very well, so I suggest explaining what indicates the acronyms of the statistics you report.

In particular, the sense of reporting AIC is a fit index for comparing different models within the same variable. There are three separate analyses, so there is no need to compare.

Author Response

  • Introduction until Aims and Hypothesis.

The article is written well; however, the authors imply readers are very experienced at the statistical level. Since hikikomori is little studied but is a pretty important topic clinically, I suggest being less synthetic in the statistical part.

Authors’ reply:

Thank you for your comment. We followed your suggestions and detailed more the statistical part of the manuscript.

  • Page 6. Attachment Style Questionnaire.

In the description of the ASQ, correct "ASQRS" in "ASQ-RS" as was done in the rest of the article.

Authors’ reply:

Thank you, we corrected the typo.

  • Page 7. 2.4 Analysis Plan

Jamovi's version always has a third number (e.g., 2.3.0). You may have used multiple versions (e.g., 2.3.22 and 2.3.26). If so, I suggest indicating, "Data analysis was conducted using the Jamovi statistical software in multiple versions 2.3.X." 

Authors’ reply:

We added the third number of Jamovi’s version to the manuscript.

  • The test is "Shapiro"-Wilk (SHA), not Saphiro-Wilk.

Authors’ reply:

We corrected the typo.

  • If you use Kendall's Tau-B correlations, why use metric analyzes after?

Authors’ reply: To explore the relationships between our variables, we used Kendall’s Tau-B correlations as our data were not normally distributed, and our sample was not large. These two aspects make us decide to use Kendall’s Tau-B correlations, which can be used for numeric variables in not large samples. Nevertheless, as our participants showed high levels of psychological distress, significantly above the scale’s cut-off, we were interested in evaluating the differences between participants having low/medium psychological distress and high/extremely high psychological distress and the role of attachment and environmental sensitivity. For this reason, we categorized depression, anxiety, and interpersonal sensitivity into two categories and ran logistic regression models, reporting goodness of fit results.

  • In Table 1, it needs to be clarified what is in the first column. I would suggest moving "n = 72" to the caption and using "Habits" as a column title; in fact, I believe that the content of the column depends on "asked to range their habits from always to never." In addition, in Table 2 (where you have space), add the relative words to EOE, AES, and LST.

Authors’ reply:

Thank you, we modified Tables 1 and 2 following your suggestions.

  • In the explanatory text of Table 3 (but also in all results), I suggest writing the names of the variables used with the capital initial (e.g., "Interpersonal sensitivity") to help connect the text with the contents of the tables.

Authors’ reply:

Thank you. We modified Table 3 and all the results accordingly.

  • I suggest indicating the type of correlation (Tau-b) in the table's title.

In Table 3, I suggest adding the size (as in Table 2) and expanding the names of EOE, AES, and LST. Finally, add a legend for INT, DEP, and ANX.

Authors’ reply:

We followed your comment and added the type of correlation in the Table’s title. We expanded the size of Table 3 and added the extended names for all the scales reported in the Table.

  • Pages 9-10.

I understand you have made three logistical regressions, one for each dependent variable. Have you considered also using Bonferroni's correction (alfa/3)?

In logistical regression, you use both age and gender, with non-significant results. It would be appropriate to indicate at least the probability of these non-significant results (for future meta-analysis) and specify if they remained in the model.

Authors’ reply:

We used Bonferroni’s correction to protect from Type I errors and specified it in the text (see page 8). As for gender and age, we added their coefficients and results in Table 5.

I also suggest separating the two tables that are currently Table 4.

Authors’ reply:

Following your suggestions, we split Table 4 into two: Table 4 was renamed “Goodness of fit”, while Table 5 was named “Binomial logistic regression”.

  • In addition, only some of the article's readers know the logistical regression very well, so I suggest explaining what indicates the acronyms of the statistics you report.

Authors’ reply:

Thank you for your suggestion. We added a note in Table 5 explicating the acronyms of the reported statistics.

  • In particular, the sense of reporting AIC is a fit index for comparing different models within the same variable. There are three separate analyses, so there is no need to compare.

Authors’ reply:

Thank you for your suggestion, we deleted the AIC index from the manuscript.

Reviewer 3 Report

After an adequate and complete introduction the authors expose the procedure, it is important to include the ethical considerations that were taken with the minor participants.

In the analysis plan a separation is made between high/extremely high levels of psychological distress and low or medium levels of psychological distress. It is very important that the authors explain how they have made this separation (using the median, the percentiles...) because most of the subsequent results depend on this separation criterion.

Likewise, it is interesting that the authors justify the choice of binary logistic regression as a data analysis technique since, having quantitative measures of the dependent variables, wouldn't it have been preferable to use a conventional regression?

It is known that the data were collected before the pandemic produced by COVID19 but in the discussion it would be of interest to make some indication of how the pandemic could influence the hikikomori or affect (modulate) the possible results of the research. It can even be proposed as a future study.

Author Response

After an adequate and complete introduction the authors expose the procedure, it is important to include the ethical considerations that were taken with the minor participants. 

Authors’ reply:

Thank you for this comment. Within the procedure paragraph of the manuscript, we better explained how minor-age participants were invited and involved in the study. We have collected different informed consent forms, one for under-age and another for their parents, as requested by Italian law and prescribed by ethical guidelines for psychologists. In the informed consent devoted to under-age participants, we adapted the description of the study and made it as accessible as possible to let the potential participants freely understand whether they were actually interested in participating in the research. Following your suggestion, we better detailed the procedure paragraph of the manuscript (page 6) to aware the reader about the ethical considerations we took into account in conducting the study.

In the analysis plan a separation is made between high/extremely high levels of psychological distress and low or medium levels of psychological distress. It is very important that the authors explain how they have made this separation (using the median, the percentiles...) because most of the subsequent results depend on this separation criterion. 

Authors’ reply:

Following your suggestion, we added an explanation about the distinction between levels of psychological distress (see page 8: “To categorize participants’ answers, we used the Hopkins Symptom Checklist (SCL-90-R) scale’s clinical cut-off: a score between 45 and 54 was categorized as a low or medium level of psychological distress, while a score between 55 and 75 as a high or extremely high level”).

Likewise, it is interesting that the authors justify the choice of binary logistic regression as a data analysis technique since, having quantitative measures of the dependent variables, wouldn't it have been preferable to use a conventional regression?

Authors’ reply:

We chose a binary logistic regression because our participants expressed high levels of psychological distress, with scores above the cut-off of SCL. As we did not have participants under this cut-off, we believed that taking the mean and running a linear regression would not have been fair to the data. Instead, categorizing our participants in low-medium and high-extremely high psychological distress categories allows us to understand differences among participants having less or more psychological distress and evaluate the relationships between all the study variables.

It is known that the data were collected before the pandemic produced by COVID19 but in the discussion it would be of interest to make some indication of how the pandemic could influence the hikikomori or affect (modulate) the possible results of the research. It can even be proposed as a future study.

Authors’ reply:

We added a part in the discussion (see page 12) highlighting that our study was conducted before the COVID-19 pandemic, saying that we can imagine that suffering from the hikikomori condition has been a risk factor during the pandemic and proposing future studies focused on the consequences of COVID-19 on hikikomori sufferers.
